# Nuclear Response to Second-Order Isospin Probes in Connection to Double Beta Decay

**Francesco Cappuzzello** [1,2,*,†] **and Manuela Cavallaro** [2,†]

1   Dipartimento di Fisica e Astronomia "Ettore Majorana", Università di Catania, 95123 Catania, Italy
2   Istituto Nazionale di Fisica Nucleare, Laboratori Nazionali del Sud, 95123 Catania, Italy; manuela.cavallaro@lns.infn.it
*   Correspondence: cappuzzello@lns.infn.it
†   On behalf of the NUMEN Collaboration.

**Abstract:** One of the key ingredients needed to extract quantitative information on neutrino absolute mass scale from the possible measurement of the neutrinoless double-beta ($0\nu\beta\beta$) decay half-lives is the nuclear matrix element (NME) characterizing such transitions. NMEs are not physical observables and can only be deduced by theoretical calculations. However, since the atomic nuclei involved in the decay are many-body systems, only approximated values are available to date. In addition, the value of the coupling constants to be used for the weak interaction vertices is still an open question, which introduces a further indetermination in the calculations of NMEs. Several experimental approaches were developed in the years with the aim of providing useful information to further constrain the theory. Here we give an overview of the role of charge exchange reactions in this scenario, focusing on second-order processes, namely the double charge exchange (DCE) reactions.

**Keywords:** charge exchange reactions; double beta decay; nuclear matrix elements

## 1. Introduction

Neutrinoless double beta decay ($0\nu\beta\beta$) is a hypothetic class of nuclear processes where a parent nucleus is transformed into an isobar daughter differing by two unit charges, and two electrons (or positrons) are emitted. Double electron capture and single electron capture connected to a positron emission are alternative mechanisms included in the $0\nu\beta\beta$ decay. Despite still not observed, these phenomena are nowadays strongly investigated since, if discovered in the experiments, they would allow to directly determine the Majorana nature of neutrino and to unveil that the total lepton number is not always conserved in nature [1]. Moreover, the neutrino effective mass would be extracted from decay rate measurements, with foreseen sensitivity to normal or inverted hierarchy scenarios in the neutrino mass distribution. Presently, this physics case is leading the research "beyond the standard model" and could open new perspectives toward a grand unified theory of fundamental interactions and contribute to explain the matter–antimatter asymmetry observed in the Universe.

Double beta decay processes occur in atomic nuclei, thus making nuclear structure issues essential for a proper description of such phenomena. The $0\nu\beta\beta$ decay rate $[T_{1/2}]^{-1}$ is typically expressed as the product of three main factors: (i) a phase-space parameter $G_{0\nu}$, (ii) a nuclear matrix element (NME) $M_{0\nu}$ and (iii) a function $f(m_i, U_{ei}, \zeta_i)$ of the neutrino masses $m_i$, the mixing coefficients $U_{ei}$ and the Majorana phases $\zeta_i$. Thus, if the NMEs are established with sufficient precision, the $f(m_i, U_{ei}, \zeta_i)$ factor, containing physics beyond the standard model, can be accessed from $0\nu\beta\beta$ decay rate measurements [2].

The NMEs are presently evaluated from theoretical calculations based on different models of the many-body nuclear wave function, e.g., proton–neutron quasi-particle random phase approximation (pn-QRPA), interacting boson model (IBM), interacting shell-model (ISM), ab-initio, energy density

functional (EDF) [3–15]. All these approaches differ in the truncation strategy adopted for the nuclear wave functions. This many-body problem ideally defined in the full Hilbert space is projected into a limited model subspace, where actual calculations can be performed. The purpose is to keep in the model space, based on reasonable physical arguments, the relevant degrees of freedom which allow a satisfactory description of the problem, leaving out unnecessary components of the wave functions. However, this condition cannot be rigorously demonstrated in advance and need to be checked by comparison with adequate experimental data. Ab-initio calculations, which are developing prodigiously, are less affected by truncation issues despite the fact that their application toward heavy nuclei is still not well established [16]. Presently NMEs calculated through the different models differ from each other within a factor of about three [14], which is still not satisfactory for the purposes of this research item. We can argue that even in the case of convergence of the calculated NMEs, the presence in all models of common assumptions could generate unknown overall systematic uncertainties [17].

A pertinent example is the actual value of the coupling constants to be used for the weak interaction operators within the nuclei. In particular, the need for renormalization of the axial-vector component $g_A$ to account for missing aspects of nuclear models is today a matter of lively discussion pointing out the limitation of the many-body nuclear calculations [14,18–26].

High precision experimental information from two-neutrino double beta decay ($2\nu\beta\beta$) [2], ordinary muon capture [27,28] nucleon transfer reactions [29–31] gamma-ray spectroscopy [32–36] double gamma decay [37] single charge exchange (SCE) [38,39] and double charge exchange (DCE) reactions [40] are or could be used to constrain the calculations. Recently, "multi-messenger" approaches, where several observables are determined all together in the same experiment and analyzed in a consistent theoretical framework, have also attracted interest [41].

In this framework, key information could be provided by the study of DCE reactions, which change the nuclear charge by two units keeping the mass number unvaried, in analogy to the $\beta\beta$-decay. An interesting observable is the DCE cross-section for the transition to the ground state of the residual nucleus, as the corresponding NME could be connected to the NME of $0\nu\beta\beta$ decay linking the same initial and final states [40]. The spectral shape of DCE cross-section has also attracted recent interest due to its connection to the nuclear response to double Fermi and double Gamow–Teller operators, giving access to the experimental scrutiny of model-independent sum-rules and providing supplementary hints to NME of $0\nu\beta\beta$ decay [42,43].

Recently, experimental DCE reaction cross-sections have become accessible in heavy-ion-induced collisions at bombarding energies above the Coulomb barrier, thanks to the use of modern high resolution and large acceptance magnetic spectrometers [40,44–46]. The experimental access to high-resolution energy spectra and accurate absolute cross-sections at very forward angles is crucial to identify the transitions of interest [47].

Based on these results, new projects have been recently proposed in Italy and Japan [41,48,49] investigating the nuclear response to DCE reactions for all the isotopes candidate for $0\nu\beta\beta$ decay.

The present review aims at presenting single and double charge exchange reaction studies in view of the connection of the nuclear response explored by these probes with $\beta$ and $\beta\beta$ decays. In Section 2, the role of single charge exchange reactions and single $\beta$ decays is discussed as first-order isospin probes. In Section 3, an extension to second-order is given through a description of DCE reactions and $\beta\beta$ decays. Section 4 focuses on the renormalization of the spin isospin coupling constants needed in the calculations.

## 2. First-Order Isospin Probes: Single Charge Exchange Reactions

SCE reactions are widely used tools for the selective investigation of the response of nuclei to neutron–proton symmetry. In an SCE reaction denoted by $A(a,b)B$, a neutron (proton) of the target transforms into a proton (neutron), $\Delta Z_A = \mp 1$, $\Delta N_A = \pm 1$, without changing the mass number, while the opposite transition, $\Delta Z_a = \pm 1$, $\Delta N_a = \mp 1$, simultaneously affects the projectile. Using the isospin degree of freedom, SCE reactions probe, at two-body level, the isovector excitations induced by

a combination of the isospin rising and lowering operators $\tau_{a\pm}\tau_{A\mp}$ acting on a nucleon in the projectile *a* and the target *A*, respectively. The monopole response $\Delta L = 0$ is particularly interesting since the associated $\sigma\tau$ operator is formally analogous to the Gamow–Teller (GT) one acting in the β-decay.

Several studies of SCE reactions have been published since the late fifties. Important reviews of the early achievements are found in the reports by Alford and Spicer [50], for a survey of the experimental explorations with light projectiles and by Osterfeld [51] for a detailed description of the theoretical aspects. Another relevant paper was published by Taddeucci et al. [52], proposing a useful factorization of the (p,n) and (n,p) cross-section into a reaction factor, named unit cross-section, and a matrix element connected to the nuclear structure overlap. More recently, the attention has been pointed out to SCE reactions induced by heavy projectiles, mainly discussed in a recent review from Lenske et al. [53].

### 2.1. Connection of Single Charge Exchange Reactions with β Decay

SCE reactions are driven by the strong interaction, with the exchange of isovector mesons, the lightest of which being the pions $\pi$. At low momentum transfer compared to the $\pi$ mass, the SCE dynamics is weakly influenced by the meson form factors and a simplified description in terms of smoothly energy-dependent coupling factors is allowed. This approach is like the one typically adopted for the weak interaction, where constant coupling factors $g_v$ and $g_A$ control the isospin and spin-isospin operators. As a consequence, SCE reaction studies are complementary to β-decay ones as far as the nuclear response to isovector probes are concerned.

A typical example is the investigation of the GT nuclear transitions. In this case, the study of the isovector monopole response ($\Delta J^\pi = 1^+$, $\Delta L = 0$; $\Delta\sigma = 1$; $\Delta\tau = 1$) by β decay is only possible within a reduced accessible energy window, but this is not the case for SCE reactions. Since the $\sigma\tau$ operator is not a symmetry for nuclear systems, the associated GT strength is broadly fragmented over many states corresponding to different excitation energies in the region of the Gamow–Teller Resonance (GTR) [54,55]. The GT distribution is a fingerprint of the nucleus, reflecting in detail its peculiar many-body nature. Therefore, the nuclear physics community has continuously put efforts into the exploration of GT strength.

When SCE reactions are used to investigate GT modes in nuclei, particular care should be paid to keep the momentum transfer as small as possible in order to filter out $\Delta L \neq 0$ components in the collision or easily unfold them in the data analysis. In addition, at vanishing momentum the tensor components of the isovector nucleon-nucleon interaction ($\Delta J^\pi = 1^+$, $\Delta L = 2$; $\Delta\sigma = 1$; $\Delta\tau = 1$) give a small contribution to the observed $\Delta J^\pi = 1^+$ strength. Such condition is typically matched at incident energy above 100 MeV/u and very forward scattering angles. Under these experimental conditions, the measured cross-sections for (n,p) and (p,n) SCE reactions were found to be proportional to known β⁺ and β⁻ strengths, respectively. However, the experimental resolution did not always allow to separate all the states populated by GT transitions in the energy spectra, somewhat limiting the accuracy of these analyses. In the years, GT studies have also been performed via SCE reactions induced by heavier projectiles, such as the (d,²He), (t,³He), (⁷Li,⁷Be) (¹²C,¹²N) (¹⁸O,¹⁸F) for the β⁺-like target transitions or the (³He,t), (¹²C,¹²B) for the β⁻-like class.

From the experimental point of view, the campaign of measurements of (³He,t) reactions mainly conducted at the Grand Raiden magnetic spectrometer of Research Center for Nuclear Physics of Osaka University (RCNP) laboratories [56–58] at 140 MeV/u incident energy has led to state-of-art results mainly thanks to the zero-degree mode for the spectrometer and the high energy resolution achieved (typical full-width-half-maximum ~25 keV) from the application of the powerful dispersion matching technique. A remarkable proportionality (better than 5%) between measured cross-sections and known β⁻ strengths are reported as a general finding, at least for the less suppressed transitions, for several nuclei widely distributed in the nuclear chart. Consequently, the RCNP facility has represented an ideal tool for high-resolution GT studies, fostering tremendous progress in the field. For the transitions of β⁺ type, remarkable results have been obtained by the (d,²He) studies at KVI-Center for Advanced

Radiation Technology, University of Groningen and RIKEN facilities [59–64]. The detection of the two protons decaying from $^2$He with high efficiency has guaranteed an overall energy resolution of about 100 keV in the missing mass spectra. The results of these experiments show that at center-of-mass detection angle for the $^2$He system around zero degrees and at 100 MeV/u bombarding energy, a close roportionality is found between NMEs extracted from SCE cross-sections and NMEs extracted from $\beta^+$ and EC studies.

An interesting application of high-resolution ($^3$He,t) and (d,$^2$He) studies is to map the GT response of specific nuclei, which represent the intermediate systems in $2\nu\beta\beta$ decay. The GT response of the even-even parent and daughter nucleus populating the odd-odd intermediate system is separately explored. The $1^+$ states of the intermediate system, which are significantly populated in both SCE reactions, are inferred to give the main contribution to the $2\nu\beta\beta$. A drawback is that the experiments access only the transition probabilities to individual $1^+$ states, while the $2\nu\beta\beta$ calculations require the amplitudes with the proper phase since their coherent sum is needed to determine the decay rate. The easiest case is when a single $1^+$ state is dominant in the intermediate state, as this prevents any coherent sum of amplitudes in this case. Approximate schemes have also been successfully adopted for $1^+$ transitions close to the Fermi level [65]. Recently, the ($^3$He,t) reaction has been used to map also the $2^-$ state distribution, opening a new promising way to provide accurate information for $0\nu\beta\beta$ NME [66].

## 2.2. Heavy-Ion Single Charge Exchange Reactions

When moving to heavier projectiles, the complex many-body nature of the involved nuclei should be taken into account as much as possible in the analyses of the SCE measured cross-section [67], and this represents an issue to account for. The projectile-target nucleus-nucleus potential needs to be accurately modeled both in the entrance (initial state interaction, ISI) and exit (final state interaction, FSI) channel. Due to the strong absorption of the incoming waves in the inner part of heavy nuclei, the direct quasi-elastic SCE reactions are localized in the nuclear surfaces of the colliding systems. This aspect of the heavy-ion reaction mechanism plays a crucial role, allowing a strong simplification of the full many-body reaction problem. Consequently, the direct reactions as SCE can be treated as perturbations of the direct elastic scattering flux, this latter described by a proper choice of the nucleus-nucleus average optical potential. Recently, ISI and FSI potentials extracted by double folding integrals of the densities of the colliding systems with nucleon-nucleon interaction have been reliably used for detailed analyses of reaction observables [68–73]. Further improvements are obtained when measurements of elastic scattering cross-sections of the projectile-target system at the same incident energy of the SCE reaction are available to constrain the calculations. In this way, the experimental SCE cross-sections may give direct access to nuclear structure features connected to the nuclear response to two-body operators, as discussed above for the GT case as well as for higher multipolarities.

In a projectile-target nuclear collision, other quasi-elastic mechanisms are allowed. As an example, multi-nucleon transfer reactions, featuring nucleon exchange among the colliding partners, could contribute to the observed SCE cross-section. In particular, the transfer of a neutron/proton from the target to the projectile (pick-up) followed by the transfer of a proton/neutron from the projectile to the target (stripping) is a composite two-step mechanism feeding the SCE outgoing channel. Since this process is indistinguishable from the direct one-step SCE mechanism mediated by two-body nucleon-nucleon interaction, an interference is expected in the reaction observables. The two-step SCE mechanism is sensitive to the nucleon-nucleus mean field potential and does not probe the nucleon-nucleon interactions stimulating the Fermi and GT response of nuclei. It is thus an unwanted complication that should be accounted for in the data analysis and minimized by a proper set of experimental conditions [74].

From the theory side, the competition between one- and two-step reaction mechanism in SCE has been scrutinized so far with major progress achieved by the development of microscopic approaches for the interpretation of the dataAn updated view of the present status of this research field is reported

in reference [53] As a general trend, the two-step mechanisms give decreasing contributions when the incident energy raises far above the Coulomb barrier. Interesting results have been reported in ($^{12}$C,$^{12}$B) [75], ($^{12}$C,$^{12}$ N), ($^{13}$C,$^{13}$ N) [76] and in ($^7$Li,$^7$Be) [77–82] reactions on several targets at incident energies from 5 to 70 MeV/u. In references [79,81] GT matrix elements were extracted from ($^7$Li,$^7$Be$_{gs}$(3/2$^-$)) and ($^7$Li,$^7$Be$_{0.43}$ MeV(1/2$^-$)) measured cross-sections for isolated transitions on light neutron-rich nuclei such as $^{11}$Be, $^{12}$B, $^{15}$C and $^{19}$O at about 8 MeV/u incident energy. A good accuracy (better than 10%) is achieved when a fully consistent microscopic approach for the ISI, FSI and reaction form factors is adopted in the calculations.

In heavy-ion-induced SCE reactions, a large amount of linear and angular momentum is typically available and transferred to the final asymptotic state, even at small scattering angles. This feature is normally considered a drawback of heavy-ion-induced SCE reactions, hindering clean access to the $L = 0$ modes, namely the Fermi and GT. However, this property turns out to be useful when the goal is to probe the nuclear response to the higher multipoles of the isospin (F-like) and spin-isospin (GT-like) operators. In fact, neither β-decay nor many of the light ions-induced SCE reactions are sensitive to high multipolarities. Nowadays, growing attention is paid to such nuclear structure features thanks to their implications in 0νββ decay NMEs [83,84], where high-order multipoles are expected to give a large contribution [85]. Thus, the use of heavy-ion-induced SCE reactions as spectroscopic tools for isospin modes has recently regained favor, outlining the interest in developing suitable experimental techniques and advanced theoretical methods for a detailed interpretation of the data [67].

## 3. Second-Order Isospin Probes: Double Charge Exchange Reactions

DCE reactions are nuclear processes induced by a projectile on a target, in which two neutrons (protons) of the target are converted into two protons (neutrons), $\Delta Z_A = \mp 2$, $\Delta N_A = \pm 2$, with the opposite transition, simultaneously occurring in the projectile, $\Delta Z_a = \pm 2$, $\Delta N_a = \mp 2$. As a consequence, the mass number of projectile and target remains unchanged. DCE reactions can be, in principle, used as a probe for selective investigation of the response of nuclear states to two-neutron/two-proton symmetry. DCE reactions probe, at four-body level, the double isovector excitations generated by a combination of the isospin rising and lowering operators $\tau_{a\pm}\tau_{a\pm}\tau_{A\mp}\tau_{A\mp}$ acting on two nucleons in the projectile and the target, respectively.

DCE transitions in the target nucleus can also be induced by accelerated pion beams, being denoted as ($\pi^+$,$\pi^-$) or ($\pi^-$,$\pi^+$) reactions. Furthermore, double beta (ββ) decay processes induce the same transition in the parent nucleus, although allowed only for positive $Q$-value. As for SCE, DCE reactions probe nuclear response to the isospin degree of freedom, despite in DCE selects second-order effects.

Here we briefly recall some relevant features of known nuclear processes driven by second-order isospin operators, emphasizing similarities and differences with the hypothetic 0νββ decay process.

In 2νββ decay, mediated by the heavy gauge bosons of the weak interaction, the GT operator acts in two independent steps, each time exchanging with the nuclear states a vanishing amount of momentum. On the other hand, the 0νββ decay is connected to the nuclear response to two-body isospin operators, which carry a sizable amount of momentum, broadly distributed around 0.5 fm$^{-1}$, and consequently excite virtual states up to high multipolarities [85]. Therefore, despite 2νββ and 0νββ decays are both weak processes, connecting the same states in the parent and daughter nuclei, they map different regions of the involved nuclear wave functions in the momentum space. The connection between the two phenomena is thus not strong enough for a safe extrapolation of 0νββ NMEs from 2νββ NMEs.

Other second-order processes of interest are the ($\pi^+$,$\pi^-$) or ($\pi^-$,$\pi^+$) pion-induced DCE reactions, in which the isospin components of the strong interaction act twice in the sequential interaction of two independent nucleons with the π fields. In the first step, n ($\pi^+$,$\pi^0$) p, the charged incident pion is converted to a neutral one; in the second step, n ($\pi^0$,$\pi^-$) p, the neutral pion is converted to a charged one. Contextually two neutrons of the initial target nucleus are converted into two protons of the final residual system, similarly for DCE induced by negative pions, where a p ($\pi^-$,$\pi^0$) n step is

followed by a p $(\pi^0,\pi^+)$n one, with the transformation of two protons in two neutrons in the nucleus. Extensive exploration of $(\pi^+,\pi^-)$ reactions was performed in the 80's leading to the discovery of second-order collective excitations as the double isobaric analog state (DIAS) and the isobaric analog state built on the top of the giant dipole resonance (GDR-IAS). The Double Gamow–Teller (DGT) was instead missed in the energy spectra. This fact was attributed to the spin-less nature of pions, making spin-isospin nuclear responses not directly accessible and thus difficult to be observed in pion-induced reactions. Johnson et al. have outlined the role of the $\Delta_{33}$ (1232) resonance in pion-induced DCE reactions [86]. The $(\pi^+,\pi^-)$ process is described as a two-nucleon mechanism through the excitation and decay of intermediate $\Delta_{33}$(1232) resonances. Auerbach et al. have deeply investigated the relevant nuclear structure features in $(\pi^+,\pi^-)$ reactions [87–89], emphasizing the central role of nucleon–nucleon correlations. Recently, Lenske et al. [53] have pointed out that correlation-driven processes are not specific for pion-induced DCE and can also manifest in other hadronic reactions. In addition, since nucleon-nucleon correlations influence $0\nu\beta\beta$ dynamics, the study of such correlations in DCE reactions may provide key information.

However, the effect of such correlations can only be observed if rank-2 isotensor processes are allowed, thus excluding processes involving isolated nucleons. Since two-proton and two-neutron systems are unbound, the projectiles for DCE must have at least a mass number equal to three; thus, the lightest allowed ones are tritons or $^3$He. However, in this case, the reactions of interest, the (t,3p) or $(^3$He,3n), are very challenging from the experimental point of view since one should detect with high efficiency the three emitted protons or neutrons in coincidence in order to reconstruct the DCE ejectile momentum. When heavier projectiles are considered, the experiments are still rather demanding. If one requires that the final ejectile is in a bound state, in order to easily identify the DCE channel in the experiments, no light nucleus can be practically used as a projectile and $^{12}$C, $^{18}$O, $^{20}$Ne or heavier projectiles are needed. Pioneering explorations of the heavy-ion-induced DCE were performed at Berkeley, Institut de Physique Nucléaire d'Orsay, Australian National University-Pelletron, National Superconducting Cyclotron Laboratory—Michigan State University, Los Alamos laboratories [90–94]. These studies focused on the $(^{14}$C,$^{14}$O), $(^{18}$O,$^{18}$Ne) and $(^{18}$O,$^{18}$C) reactions at energies above the Coulomb barrier, often with the main purpose of measuring the mass of neutron-rich isotopes by reaction Q-value measurements. However, these experiments were not conclusive for spectroscopic purposes, mainly because of the poor statistical significance of the few DCE collected events; thus, no further DCE measurement was performed for a long time. Furthermore, the development of theories to investigate the DCE reaction mechanism [95,96] soon slowed down, and the field was almost abandoned for many years.

*DCE Reactions and 0νββ Decays*

Recently, DCE studies have raised an increasing interest, also because of their possible connection to double beta decay issues. New reactions have been explored at RIKEN and RCNP at energies between 80 and 200 MeV/u. The $(^8$He,$^8$Be) reaction was adopted to search for the tetra-neutron (4n) resonances by the $^4$He$(^8$He,$^8$Be)4n at 186 MeV/u [97]. The $(^{11}$B,$^{11}$Li) [98] and the $(^{12}$C,$^{12}$Be) [99] were investigated to search for the Double Gamow–Teller Giant Resonance (DGTGR) and provide quantitative information about the DGT sum-rule, of interest for modern nuclear structure theories [100]. Another DCE reaction, the $(^{20}$Ne,$^{20}$O), has been introduced for the first time by the NUMEN (NUclear Matrix Elements for Neutrinoless double beta decay) and NURE (NUclear REactions for neutrinoless double beta decay) projects [41,48] with the aim to probe nuclear response to a $\beta^-\beta^-$-like transition. In addition, renewed use of the $(^{18}$O,$^{18}$Ne) reaction in upgraded experimental conditions has allowed achieving important results. The $^{40}$Ca$(^{18}$O,$^{18}$Ne)$^{40}$Ar DCE reaction, studied in ref. [40] at 15 MeV/u at the MAGNEX facility at Istituto Nazionale di Fisica Nucleare-Laboratori Nazionali del Sud (INFN-LNS) [44,45,47,101,102] has shown that high mass, angular and energy resolution energy spectra and accurate absolute cross-sections are at reach, even at very forward angles including zero-degree. Moreover, in the same

paper, a schematic analysis of the measured cross-sections has demonstrated that DCE matrix elements can be extracted from the data and compared with nuclear structure calculations.

An interesting advantage of the new proposed projects on DCE reactions, such as NUMEN, is to potentially cover all the isotopes of interest for $0\nu\beta\beta$ decay. In principle, specific experimental issues, as for example, the technology to produce nuclear enriched isotopic targets or the energy resolution necessary to separate the transition to ground state, could make some double beta emitters easier to be investigated. However, such aspects can be reasonably managed to make all the $0\nu\beta\beta$ emitters experimentally accessible via DCE [41].

A general experimental challenge of heavy-ion-induced DCE is the very small cross-section characterizing such processes, demanding for high beam current and requiring a significant upgrade of the present facilities. Such pioneering studies have indeed motivated the ongoing major upgrades of the INFN-LNS laboratory infrastructure, in view of providing very intense heavy-ion beams for DCE experiments and also for the whole nuclear physics community [41,103]. First results from the upgraded facility are expected in a 3–4 years time horizon.

Resembling the case of heavy-ion-induced SCE reactions, also for DCE, an important issue is to give a quantitative evaluation of the contribution due to multi-nucleon transfer processes with respect to the "direct" meson-exchange one. In the case of DCE, the transfer effects are of 4[th] order in the nucleon-nucleus potential since four nucleons are involved; two protons (neutrons) are stripped from the projectile, and two neutrons (protons) are picked-up from the target. In ref. [40] it was shown that the contribution of multi-nucleon transfer is negligible (less than 1%) for the $^{40}$Ca($^{18}$O,$^{18}$Ne)$^{40}$Ar reaction under the experimental conditions set for the measurement at INFN-LNS. Similar results are found in the preliminary analysis of the other explored cases [74]. The leading DCE reaction mechanism is thus mainly driven by the nucleon-nucleon isovector interaction, with negligible contribution from the exchange of nucleons between projectile and target. A useful way to model the DCE direct process is by means of the exchange of two charged $\pi$ or $\rho$ mesons between two nucleons in the projectile and two nucleons in the target. A pertinent open question is whether the two mesons are exchanged independently of each other in analogy to $2\nu\beta\beta$-decays [104] or in a correlated way, as in the $0\nu\beta\beta$-decays [53,105]. Answering this question is relevant for the connection of the nuclear response probed by DCE reaction and $0\nu\beta\beta$ decay. In addition, this aspect could have an impact on nuclear reaction theory since it could indicate a new way to access selective features of nucleon-nucleon short-range correlations [53].

The recent availability of high-quality experimental data on DCE reaction observables raises the question of how they can be profitably used toward the experimental access to $0\nu\beta\beta$ decay NMEs. NMEs from DCE reactions and $0\nu\beta\beta$ decay requires the same degree of complexity for the nuclear structure model, with the advantage for DCE to be "accessible" in the laboratory under controlled conditions. In ref. [40] it has been pointed out that, although the DCE and $0\nu\beta\beta$ decay processes are mediated by different interactions, there are a variety of important similarities among them:

- The initial and final states (parent and daughter) of the $0\nu\beta\beta$ decay are the same as the initial and final states (target and residual nuclei) in the DCE reaction;
- Both operators present short-range Fermi, Gamow–Teller and rank-2 tensor components, even if with different relative weights, depending in principle on the incident energy in the reaction case. The DCE experiments at different beam energies could give information on the individual contribution of each component;
- In both processes, a large linear momentum (~100 MeV/c) is available in the virtual intermediate channel [106]. It is worth to underline that other processes such as single $\beta$ decay, $2\nu\beta\beta$ decay, SCE reactions induced by light ions are characterized by small momentum transfer, so they cannot probe this feature [107]. The recently proposed $\mu$-capture experiments [108,109] could represent interesting developments in this context;
- In both cases, the processes require non-local operators acting on the same pairs of nucleons;

- Both transitions take place in the same nuclear medium. Since effects due to the presence of the medium are expected in both cases, DCE experimental data could give a helpful constraint on the theoretical determination of quenching phenomena in $0\nu\beta\beta$;
- Off-shell propagation through virtual intermediate nuclear states features both cases. Since the virtual states do not represent asymptotic channels, their energies are not well defined as those (measurable) at stationary conditions [110].

In ref. [105], a useful factorization formula of DCE cross-section in a nuclear structure term and a reaction part was suggested within the semi-classical eikonal approximation for the reaction, at least for the $0^+$ to $0^+$ transition from the ground state of an even–even parent to the ground state of the even-even daughter nucleus. This factorization is found to be possible for the differential cross-section at $\theta = 0$, where the transition matrix elements can be written as the sum of double Gamow–Teller and double Fermi-type parts and that they can both be further factorized in terms of target and projectile NMEs.

In ref. [104], this factorization for $0^+$ to $0^+$ transitions was proven to hold in a more advanced nuclear reaction model based on a fully quantum mechanical distorted wave two-step approach at vanishing momentum transfer. These conditions are verified at a very forward scattering angle for heavy-ion-induced DCE reactions. In the same work, a similarity between two-step sequential component of DCE cross-section and $2\nu\beta\beta$ decay is emphasized, despite DCE reactions cover a larger spectrum of momentum transfer. A comparison of the calculation with DCE differential cross-section data gives promising results in terms of the description of an absolute cross-section. However, room is left for additional contributions from correlated one-step DCE mechanism in order to explain the shape of the angular distribution at very forward angles. This aspect is further deepened in ref. [53], where the correlated one-step DCE mechanism, called "Majorana DCE mechanism", is calculated in a fully microscopic approach and found to be essential in order to reproduce the experimental data. The Majorana DCE mechanism is indeed very interesting, as it is driven by short-range nucleon–nucleon correlations, similarly to $0\nu\beta\beta$ decay.

Such pioneering works, whose application is still limited to few available data, are indeed very encouraging and support a deeper investigation of both the experimental and the theoretical features.

In Refs. [42] and [111], specific aspects of this analogy have been investigated, searching for additional physical observables related to $0\nu\beta\beta$ NMEs. The authors find an interesting connection between the centroid energy of DGTGR for $^{48}$Ca and the $0\nu\beta\beta$ NME feeding the ground state of the daughter nucleus, both quantities calculated within a large scale shell model framework. DGTGR cannot be accessed in a double beta decay as it sits mainly in the particle continuum portion of the energy spectrum; instead, it is in principle accessible by DCE reactions. If measured, DCE cross-section energy distribution would allow getting the associated DGT strength distribution, using, for example, recently developed techniques as that proposed by V. dos S. Ferreira et al. [43]. In addition, the DGT matrix element for the transitions to the ground state of the final nucleus and the $0\nu\beta\beta$ decay NMEs are also found to be inherently connected for several nuclei, including $\beta\beta$ emitters. Such connection holds for different calculation schemes, with the important deviation found for QRPA.

## 4. The Renormalization of the Spin-Isospin Coupling Constant

An open question in the exploration of the nuclear response to spin-isospin operators is the actual value of the coupling constants to be used in the calculations.

For the weak interaction operators, the need for a scaling factor for the axial-vector component $g_A$ is today much debated [14,18–26]. The value of $\frac{g_A}{g_V} = -1.27641(45)_{stat}(33)_{sys}$ was recently determined with an unprecedented resolution by polarized neutron decay measurements [112]. However, the use of this "free" coupling constant also for a process occurring within an atomic nucleus is not fully justified for several reasons. These reasons can be grouped in two main classes: (i) the incomplete description of nuclear many-body correlations in the adopted model [113–115] and (ii) the omission of many-nucleon weak currents (especially two-body currents) [116]. Both give a sizable influence on processes driven by the weak interaction, including $0\nu\beta\beta$, with the consequent need for an effective coupling constant

$g_A^{eff}$ which accounts, on average, for the missing physics. The actual value of the coupling constant thus depends on which of the two sources gives a larger contribution to the renormalization.

This problem resembles the historical puzzle of the "missing GT strength" that has challenged the nuclear physics community for about half a century. The striking observation is that only 50% to 70% of the strength predicted by the Ikeda sum rule (defined as the difference $(S^- - S^+)$ between the energy integrated strength for GT transitions mediated by $\sigma\tau^-$ and $\sigma\tau^+$ operators) [117], [51] is found in charge-exchange experiments [118]. Since the Ikeda sum-rule is a model-independent relation, $S^- - S^+ = 3(N - Z)$, directly derived by commutation properties of the $\sigma\tau$ operators, it should represent a benchmark for theories and experiments, making this discrepancy very disappointing. The search of the missing strength has pushed the experiments toward higher excitation energies, beyond the region of the GTR up to about 50 MeV. Exceeding this limit, the so far reported extraction of the monopole strength from the experiments is not safe and accurate [50,119]. To date, no conclusive answer has been given to this problem, despite a long and intense research activity in the field. Recently, Douma et al. [120] have reported a detailed analysis of the spectra of Sb isotopes at high excitation energy, pointing out that a careful treatment of the quasi-free component in the reaction cross-section for the ($^3$He,t) reaction could mitigate this discrepancy.

Another long-standing puzzle, likely connected to the latter, is that the GT strengths extracted from measured cross-sections of transitions to isolated low-lying states are systematically smaller than predicted with different nuclear structure models requiring a quenching factor of about 0.7 in the $\sigma\tau$ operator to reproduce the data [121–123]. Similar behavior has also been reported for GT strengths from β decays [124,125], whose experimental values were about 30% smaller than calculated by the shell model. A recent detailed analysis of the $^{113}$Cd β-decay spectral shape with different nuclear structure models indicates the need for about 20% reduction of the fourfold forbidden nonunique decay matrix element in all cases [126], thus showing persistence of quenching at high momentum transfer in β-decay. A standard way to incorporate this feature in the data analyses is to adopt a quenched coupling constant for the axial vector weak interaction and for the spin-isospin strong interactions.

Interestingly, SCE and β-decay processes, which are driven by the strong and the weak interaction, respectively, show a remarkable similarity in the $\sigma\tau$ channel. This likely indicates that something is missing in the traditional description of the nuclear response to στ operators, regardless of the nature of the probe. In this context, a recent study with a state-of-art ab-initio approach has demonstrated that, to a large extent, the explicit inclusion of two-body currents is indeed mandatory to get rid of this discrepancy, at least for β-decay [127]. In the same article, the same theoretical methods are adopted to explore the Ikeda sum-rule for $^{14}$O, $^{48}$Ca and $^{90}$Zr, showing a sizeable and mass-dependent reduction of the strength from about 20% ($^{14}$O) to about 40% ($^{90}$Zr) when two-body currents are introduced. It would be very interesting to extend this exploration to SCE, including the projectile-target interaction in the same approach. However, to our knowledge, ab-initio methods are still not sufficiently developed for SCE reactions, so to date, the role of two-body currents in SCE can be explored with less detail.

The possible need for quenching factors in the coupling constants of the weak and strong interactions in the spin-isospin sector is a key question also for second-order processes driven by isospin operators. The matter is controversial, despite the fact that an accurate answer to this issue is considered central nowadays, as these parameters enter with the 4th power in the 0νββ half-life expression and could hinder any real progress in the field.

As mentioned before, both nuclear many-body correlations [113–115] and two-body currents [116] can alter the final value of the matrix elements for $\sigma\tau$ operator. In principle, these sources act on different degrees of freedom, nucleonic for the former, non-nucleonic for the latter, so their effect can be separately scrutinized. However, the full-fledged picture needs to be consistently analyzed for a quantitative evaluation of the overall effect in the coupling constant. In ref. [14] the different role of these sources for 2νββ and 0νββ decays is discussed in terms of the very different momentum available for the two processes. A severe quenching could be caused by many-body correlations, the latter also including the short-range components of the nuclear force. This agrees with ref. [21],

showing that a sizable scaling (about a factor of two) of the axial-vector coupling constant is needed for 2νββ decay, at least within IBM2 nuclear structure approaches. On the contrary, two-body currents tend to attenuate the quenching, with a 20–30% overall effect for 2νββ decay [14]. In addition, a weak dependence on momentum available in the second-order process with a positive projection for a mild effect on 0νββ decay is foreseen [10,128].

We should mention here that such initial encouraging results of a controlled and small quenching correction factor for ββ decays have been recently questioned by a more comprehensive analysis of the leading order effects of two-body weak currents from chiral effective field theory (χEFT) [129]. Two-body effects become divergent and must be renormalized by a contact operator, the coefficient of which is completely undetermined by χEFT at present. Lattice QCD calculations could, in principle, overcome this problem, but they are missing at the present time.

We emphasize here that the above discussion relies on the intrinsic nature of the nuclear many-body problem, dressed with short-range correlations among nucleons and two-body currents connected with non-nucleonic degrees of freedom present in the atomic nuclei. The influence of these aspects is recently being analyzed in quantum Monte Carlo ab-initio also approaches for electromagnetic observables in light nuclei [130,131]. The interesting result is that two-body currents only give 2–3% correction for the electromagnetic observables, rising to 20–30% for beta decay rates. Thus, different probes are sensitive to these aspects of the nuclear response, despite at different levels. One should thus expect that such "contact operator" from χEFT can also act in DCE reactions, which thus could represent a unique source of experimentally driven information to scrutinize this effect for a second-order operator in spin-isospin.

## 5. Conclusions

An overview of the recent developments and perspectives of single and double charge exchange reactions was given, paying special attention to the connection with weak interaction-driven processes, namely β and ββ decays. Despite the differences between processes driven by different interactions, the present status of the field is promising. Historical experimental limitations plaguing DCE reactions were partly overcome and will be even more thanks to ongoing major facility upgrades, as for example, at INFN-LNS laboratory. Accurate cross-section measurements can be performed with an acceptable signal-to-noise ratio down to a few nb, allowing extracting energy spectra and angular differential distributions. In addition, recent strong progress of the theory of DCE is shedding new light on the possible quantitative connection of DCE, SCE and nucleon transfer cross-sections with nuclear structure inputs required for 0νββ decay NMEs.

**Author Contributions:** Both authors have actively contributed to the manuscript preparation and to the development of the ideas here presented. F.C. gave a specific contribution on the discussions presented in Section 4. M.C. gave a specific contribution on the organization of the manuscript. All authors have read and agreed to the published version of the manuscript.

**Funding:** This work has received funding from the European Research Council (ERC) under the European Union's Horizon 2020 research and innovation program, NURE project, grant agreement No. 714625.

**Acknowledgments:** The authors wish to acknowledge all the members of the NUMEN collaboration for fruitful discussions.

**Conflicts of Interest:** The authors declare no conflict of interest.

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
