# Peer review of "Nuclear Response to Second-Order Isospin Probes in Connection to Double Beta Decay"

_universe, doi:10.3390/universe6110217_

Round 1

Reviewer 1 Report

The paper is aimed at describing the physical principles of the Double Charge Exchange reactions (DCE), showing the connection to the determination of the Nuclear Matrix Elements (NMEs) of neutrinoless double beta decay emitters. A previous discussion on Single Charge Exchange reactions (SCE) is included too.
As the calculation of NMEs is of uppermost relevance to derive the effective mass of neutrinos from the measurement of the half-life of neutrinoless double beta decay emitters, and there are important uncertainties on those calculations, the investigation of possible validations from experiments is essential; then, in my opinion, the review has maximum interest for readers of this special issue of Universe.
I consider that the context of the studies reviewed in the paper is well presented. Contents are well organized and the previous discussion on SCE reactions is very useful for understanding later DCE reactions.
Nuclear aspects are clearly explained and the main experimental results for the different types of reactions considered are summarized.
I just propose a very minor revision before publication, mainly related to format aspects.
I list below my specific comments, indicating the line of the mansucript they refer to.

5: Should the article be considered as authored by the full NUMEN collaboration? If so, I guess that the full list of authors should be explicitly included.
15: It would be advisable to define the abbreviation 0νββ also in abstract, even if very extesively used.
16: I think "Elements" should be "Element".

SECTION 1
32: A reference to some review of experiments looking for the neutrinoless double beta decay would be recommended here to complete the introduction to the topic.
87: Among the neutrinoless double beta emitters which are presently being investigated in underground experiments, is there anyone for which the DCE reactions could give more easily information for the quantification of its NME? Or alternatively, are there isotopes for which it is more difficult to obtain the DCE reaction information?
If so, a comment on this could be in order here.
88: I would suggest including a brief summary of the goals and structure of the review at this point, as it can be helpful for the readers.

SECTION 2
138: I suggest adding the meaning of RCNP as Research Center for Nuclear Physics of Osaka University at this first appearance, to better identify facilities and institutions.
146: Similarly, I suggest completing KVI - Center for Advanced Radiation Technology, University of Groningen.

SECTION 3
272: Again, I would explain the abbreviations for IPN (L'Institut de Physique Nucléaire d'Orsay), ANU (Australian National University) and , NATIONAL SUPERCONDUCTING
CYCLOTRON LABORATORY- Michigan State University,
289: The same for NUMEN NUclear Matrix Elements for Neutrinoless double beta decay
290: The same for NURE nuclear reactions for neutrinoless double beta decay
293: The same for INFN-LNS Laboratori Nazionali del Sud
295-299: You could summarize here the quantitative information which can be extracted. It would be also very interesting to mention, if possible, the time scale to have results from the INFN-LNS facility on DCE reactions relevant for neutrinoless double beta decay.

SECTION 5
467-472: I would suggest expanding the conclusions, commenting with more detail the progress on both experiments and theory related to DCE reactions, and summarizing all the considered physical observables on DCE reactions related to neutrinoless double beta decay NMEs.

REFERENCES
[16] It would be possible to provide at least a web link to have access to the report?

Author Response

Please find attached the authors' notes to reviewer 1

Reviewer 2 Report

I would like to congratulate you on this excellent review paper, it provides a thorough and well explained overview of the topic.

Author Response

Please find attached the authors' notes to reviewer 2

Reviewer 3 Report

This is an interesting review that contains a lot of useful information.  It would be fine to publish.  But I think that the paper would be stronger if it were a little more focused.  As of now it contains no equations or figures.  If the authors could describe either the theory of  DCE or the experimental situation (or both) in a little more detail, it would help the reader.  What are the primary accomplishments, e.g., of refs. 52, 53, and 104?  Has the theoretical work led to a better understanding of data?  If so, how?  Are there DCE data that can be displayed along with theoretical predictions?

I have a few quibbles:

  1. It's not really correct to say that the "knowledge of the axial-vector component gA is today a matter of lively discussion."  gA is what it is.  The question is whether nuclear models are sufficient and, if not, whether their deficiencies can be absorbed into the use of an effective value for gA.
  2.  I don't understand the sentence "In both cases the processes are non-local with vertices localized in a pair of nucleons," which seems to contradict itself.
  3. Returning to gA, the "free" value referred to in section 4 is what should be used if the nuclear model is adequate, so it's not really OK to say that its use is not fully justified.
  4. The use of two-body currents has only really been explored within chiral EFT.
  5. I believe that the dependence of quenching on momentum transfer in two-body currents seems strong enough to expect less quenching for 0nu decay.  This would not be the case for the "weak dependence" the the authors cite.  (Weak implies a small effect)

These are all minor points and, as I said, I have no objections to the paper being published as is.  But I think it would be much stronger with a more quantitative discussion of DCE.

Author Response

Please find attached the authors' notes to reviewer 3

This manuscript is a resubmission of an earlier submission. The following is a list of the peer review reports and author responses from that submission.

Round 1

Reviewer 1 Report

Report of universe-939453
------------------------------------

The review manuscript "Search for second order response of nuclei to isospin probes and connection to double beta decay" presents a very complete report of present and past efforts in extracting nuclear structure and beta decay information from single charge exchange and double charge exchange reactions. The authors summarize the main achievements obtained with these experiments in the past, and discuss the possibility that future campaigns illuminate the understanding of neutrinoless double-beta decay (0nbb). Given the relevance of the latter process for neutrino, high-energy physics and cosmology, the current manuscript provides a very valuable review that shows the capability of nuclear reactions to shed light on rare nuclear processes such as 0nbb.

The manuscript is very well written, accessible to non experts, and almost in all instances makes reference of the appropriate literature on the field (few exceptions are in my suggestions below). It serves, in my opinion, as a very good introduction on the topic, and reviews the main findings obtained with charge exchange reactions so far. Therefore, I consider the manuscript potentially suitable for publication in Universe.

However, before I recommend the publication of the manuscript, the authors should pay special attention to the discussion, controversial as they say, of the "coupling constants to be used for the weak interaction operators within the nuclei", and two-body effects.

i) In both the abstract (line 17-18), the introduction (line 59-60) and the section on double charge exchange reactions (line 344-347) the authors should clarify, or highlight, that the "effective value of the coupling constants", or "quenching", is most likely a requirement due limitations on the many-body nuclear calculations. The best available calculations, Ref.[53] and PRC 90 024321 (2014), PRC 102 025501 (2020) reproduce experiment relatively well without any "quenching". Less involved calculations, such as the ones usually performed in heavy nuclei, require such "quenching" due to the approximations required in these methods. Nonetheless I very much agree that experiments such as DCE are very useful to quantify the "quenching" that a given approximate many-body method requires in a process involving large exchange momentum, such as 0nbb.

ii) While the two-body effects in Ref.[112], discussed around line 395, are formally divergent and need to be renormalized, this is not necessarily problematic in chiral effective field theory. The uneasiness is of course that the coupling associated with the contact operator that renormalizes two-body effects is not predicted by the theory, and should be calculated, for instance by lattice QCD. Nonetheless, the expectation---if chiral effective theory is to the taken as a reference---is that such term would not alter very significantly the (main) results of Ref.[112], which are in line with the "Initial encouraging results of a controlled and small quenching correction factor" of Refs.[110,111]. This extended discussion should be reflected around line 395, otherwise the reader gets the too negative impression that two-body effects in 2nbb have been very much "obscured".

In addition, I list here some other smaller additions and clarifications that the authors should consider to improve their manuscript:

a) at the beginning of the second paragraph in page 2, the authors list different experiments that could be used to constrain 0nbb matrix element calculations. In their list I miss spectroscopic studies exploring the nuclear structure of bb parent/daughter nuclei. For example, PRC 87 051305(R) (2013), PRC 87 041304(R) (2013), PRC 95 014327 (2017), PRC 99 014313 (2019) or PRC 99 054313 (2019) have explored A=76 systems, and PRC 98, 034302 (2018) studied A=136. These efforts are also worth mentioning, as they provide very useful tests of the theoretical calculations.

b) Some selected, or review references would be good to illustrate the long research referred to in the second paragraph of page 3:
"no safe and accurate extraction of the monopole strength from the experiments has been reported so far [...] despite a long and intense research activity in the field"

c) Also in page 3, I miss references regarding the quenching needed by theory calculations to reproduce GT decays, and GT strengths (Ref.[52] only covers the beta decay spectrum). Classical references are PRC 28 1343 (1983) or PRC 53 R2602 (1996) for GT decays, and PRC 50 225 (1994), PRC 75 034303 (2007) or JPS Conf. Proc. 6 030057 (2015) for GT strengths.

d) In page 3, the authors write that
"It would be very interesting to extend this exploration [of Ref.53] to SCE"
However, this has partly been done already, since Ref.[53] predicts corrections to the Ikeda Sum Rule tested in SCE reactions for several nuclei, see Fig.7 in the Supplementary Information of Ref.[53].

e) In the last bullet of the list initiated in page 7, I do not agree with the statement on
"Off-shell propagation through virtual intermediate channels"
for 0nbb decay. The virtual propagation occurs for off-shell neutrinos, while the intermediate states correspond to the nuclear level scheme of the intermediate odd-odd nucleus.

f) In line 353, the authors claim that a useful factorization formula for double charge exchange cross sections was "demonstrated" in Ref.[103]. Due to the approximations needed in the derivation of Ref.[103], it is more appropriate to say that a factorization formula was "suggested" by Ref.[103].

g) In page 8 the authors claim that
"DGT matrix element for the transitions to the ground state of the final nucleus and the 0νββ decay NMEs are also found to be inherently connected for several pf-shell nuclei"
However, I think the authors of Ref.[34] claim that the connection extends to heavier nuclei as well, reaching germanium and xenon isotopes.

In summary, this work is very well written and provides a very nice review of charge exchange reactions and their possible connection to 0nbb. Once my comments and remarks above are taken into consideration, I will recommend the manuscript for publication in Universe.

Reviewer 2 Report

Referee report on the paper Universe 939453 “Search for second order response of nuclei to isospin probes and connection to double beta decay” by F. Cappuzzello and M. Cavallaro

The paper is a review about a very recent issue in the modern physics. The problem of the estimation of the Nuclear Matrix Elements (NME), in connection with the double beta decay experiments. The authors, experts in the field, summarize the recent developments of single and double charge exchange nuclear processes, in the light of the phenomenology connected to the beta and double-beta decay processes.

The paper is well written and the list of references to explore deeply the argument is rather adequate. The argument is well described in main aspects. The only criticism (if we want to find one) is that I expected some quantitative results, as for example some figures/tables and comparisons among the achieved results.

However, I strongly suggest to publicate this review in Universe.

Reviewer 3 Report

This manuscript is almost a carbon copy of the paper entitled "Search for second order response of nuclei to isospin probes and their connection to double beta decay" written by the same authors and published in Journal of Physics: Conference Series 1610, 012003 (2020) as part of the conference proceedings of the XLIII Symposium on Nuclear Physics 2020. The paper is obviously sound and well-written, but it cannot be accepted for publication in another journal.